# Design and Testing of a Compact Optical Prism Module for Multi-Degree-of-Freedom Grating Interferometry Application

**Xinghui Li** [1,*,†], **Yaping Shi** [1,†], **Xiang Xiao** [1], **Qian Zhou** [1], **Guanhao Wu** [1,2], **Haiou Lu** [1] **and Kai Ni** [1,*]

1   Division of Advanced Manufacturing, Graduate School at Shenzhen, Tsinghua University, Tsinghua Campus, Xili University Town, Shenzhen 518055, China; cdshiyaping@gmail.com (Y.S.); xiaoxiang_x_x@126.com (X.X.); zhou.qian@sz.tsinghua.edu.cn (Q.Z.); guanhaowu@tsinghua.edu.cn (G.W.); lho16@mail.tsinghua.edu.cn (H.L.)
2   Department of Precision Instrument, Tsinghua University, Haidian District, Beijing 100084, China
*   Correspondence: li.xinghui@sz.tsinghua.edu.cn (X.L.); ni.kai@sz.tsinghua.edu.cn (K.N.)
†   These authors contributed equally to this work.

**Abstract:** In this research, a key optical component for multi-degree-of-freedom (MDOF) surface encoder was designed, fabricated and evaluated. In a MDOF grating interferometry system, there are four diffraction beams from a two-axis scale grating and reference grating, respectively. For further modulation, these beams will propagate more than 100 mm, which makes paralleling these beams necessary. In previous research, collimation lens, separate prisms and a home fabricated diffraction device by combining four separate one-axis line gratings in a glass substrate have been demonstrated. However, large power loss and assembly complicity makes these techniques less competitive. For solving this problem, this research proposed a new lens module, which is an improved type prism, quadrangular frustum pyramid (QFP) prism. The prism is designed in such a way that these four reflected beams from the grating are symmetrically incident into the prism through the upper surface, total reflected on the inner sides of the prism, and then parallel getting through the bottom surface. A prism that allows an incident beam diameter of 1 mm and four paralleling beams with a 10 mm distance between the two diffraction beams along one direction was designed, fabricated and tested. Testing results based on an entire grating interferometry system verified that the proposal in this research is greatly effective in beam paralleling in terms of less power loss and high paralleling and greatly reduces the assembly complicity, which will eventually be beneficial for grating interferometry application.

**Keywords:** surface encoder; multi-degree-of-freedom; interferometry; grating; prism

## 1. Introduction

With the development of industrial automation and intelligence, the popularization and demand of multi-axis computer numerical control (CNC) equipment in industrial production are increasing. Therefore, the multi-degree-of-freedom (MDOF) measurement technology is a key basic technology. For instance, in a MDOF CNC machine tool, the current common measurement scheme is to equip each motion axis with a separate single-axis measuring device, including linear grating ruler, linear photoelectric encoder, and so forth [1–3]. The installation of these measuring devices is usually stacked along the motion axis. This brings about the abbe error problem in the assembly process, which will have an adverse effect on the measurement accuracy [4,5].

In order to eliminate the abbe error in multi-axis measurement and improve the integration of the measurement system, measurement devices with MDOF measurement capabilities have been paid more attention. Among them, the technique schemes of two-dimensional grating rulers and the MDOF grating rulers are of great application potential [6–9]. These MDOF grating rulers, generally with usage of a two-axis planar scale grating, are able to measure not only the translational motions but the small order rotational error motions, and it can be also applied to six-DOF measurement. Compared with the traditional single-axis measuring devices, the MDOF grating ruler has a higher level integration and the abbe error could be avoided Thus, the reliability and precision of the multi-axis precision stage and measurement system can be improved effectively [10–12].

As illustrated in these prototype MDOF grating rulers, grating diffraction beams interferometry is the key technology, which means that the interference is generated by using four beams of ±1 order diffracted light of a two-dimensional grating so that to obtain the displacement distance by detecting the interference signal [6–8]. Compared with the one-dimensional interferometric grating ruler, the two-dimensional grating ruler is equivalent to integrating two sets of one-dimensional grating rulers whose working direction is orthogonal to each other, however this makes complexity of the two-dimensional grating ruler light path increase greatly. The complexity is mainly manifested in the need of the interference of four beams both from the reference grating and the scale grating parallel or perpendicular to the incident light in the system in order to produce an ideal interference signal. This means that the above eight diffracted beams need to be finely deflected, which determines the quality of the interference signals. In previous research, there are three proposed methods, collimation lens represented by Reference [6], separate prisms represented by Reference [5], and a home fabricated diffraction device by combining four separate one-axis line gratings in a glass substrate [13]. These methods meet the demand of the light path design and can be manufactured under the lab conditions. However, due to the complex structure and alignment, loss of signal intensity and the low sensitivity, these solutions are difficult to meet the condition of mass production.

Therefore, in this paper, focusing on the demand of mass production and engineering prototype, facing these challenges in current technologies, we design a new lens module for this kind of grating rulers, a single prism called a quadrangular frustum pyramid, enabling simultaneously deflecting the four diffraction beams without any power loss and easy alignment. To test the lens module, a grating ruler system is designed, constructed and experimental verification are introduced in this manuscript.

## 2. Design of the Integrated Optical Prism

Figure 1a is the schematic diagram of the principle of two-dimensional grating ruler [5]. When a laser beam is irradiated vertically on a two-dimensional grating, the ±1 order diffractive light will be generated along the two grating lines of the two-dimensional grating. The total diffraction light is four beams, denoted as $U_{X+1}$, $U_{X-1}$, $U_{Y+1}$, $U_{Y-1}$ ($U_{Y+1}$, $U_{Y-1}$ are not shown in this Figure for clarity). The angle between the diffraction light and the normal line of the grating plane is equal to the diffraction angle of the grating. As shown in Figure 1a, the beam emitted by the laser source is vertically irradiated on the reference grating and the scale grating after the beam is divided by the beam splitter. The interference of ±1 order diffractive beams from the scale grating and reference grating occurs after the beams combining in the splitter. The interference fringes are formed and can be detected by photoelectric detectors for its light intensity, and finally four interference signals are obtained. After that, the measured displacement can be solved according to the detected interference signal through the corresponding data processing algorithm.

As illustrated in previous research [5,6,13], further phase delay and amplitude division of these signals are required so that the direct current components of the interference signals can be eliminated and the motion direction can be distinguished. Thus, more optics are added into this optical layout and this results the light path increasing to more than 100 mm [14]. This makes paralleling these diffraction beams necessary. As mentioned in the first section of this paper, Figure 1b–d illustrate these three present optical designs. As shown in Figure 1b, a collimation lens was employed. The diffraction

beams were refracted. This is suitable for large grating period, generally a 10 micrometer order. Under such a condition, the diffraction angle from gratings are relatively small, and the collimation effect can be ensured [6]. However, a large grating period will result in a relatively poor resolution. A high resolution requires a small grating period. Correspondingly a large diffraction angle makes using the collimation difficult. The second trial is that of the triangular prism refraction method, as shown in Figure 1c. Similarly, a triangular prism can be replaced with a mirror. Both methods need the diffractive light deflection mirror to be precisely calibrated so that the reference and the scale grating diffractive light will be able to interfere after transmission to the detector. It was found in the trial production of the actual test prototype [5,12], the process of assembly and adjusting the diffractive light deflection prism is extremely complicated and difficult for the mechanical assembly stress and the influence of such factors as poor precision stability of the optical path. In order to avoid the problem that diffractive light deflection prisms need to be adjusted separately, a structure using transmission grating for diffraction light deflection is proposed, as shown in Figure 1d [13,15,16]. In this structure, the transmission grating is actually four one-dimensional transmission gratings combined. This scheme greatly reduces the difficulty of assembling and debugging optical circuits by employing the integrating transmission grating, which is used as a diffractive light deflection device. However, limiting the transmission grating diffraction efficiency, the plan will greatly reduce the intensity of the diffractive light. And it is not conducive to improve the signal-to-noise ratio (SNR) in subsequent signal processing, which leads to reduce the measuring accuracy of the grating ruler system.

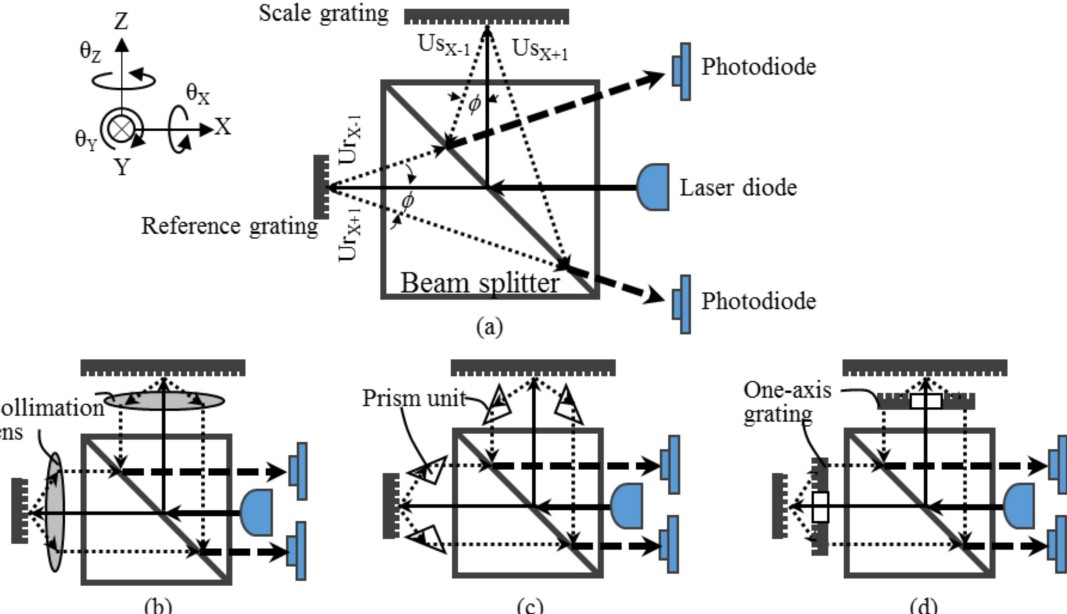

**Figure 1.** (**a**) Principle of a two-grating encoder; (**b**) diagram of employing a collimation lens to parallel diffracted beams; (**c**) diagram of employing prims to parallel the diffracted beams; (**d**) diagram of employing gratings to parallel the diffracted beams.

In order to overcome the above problems, an integrated grating diffractive light deflection prism design is proposed. The design takes the advantages of prism deflection that does not reduce the beam intensity and transmission grating high integration, providing a high precision, high brightness and easy to assemble and debug solution for reference grating and scale grating diffractive light deflection. The effect of the integrated prism and the deflection of four ±1 order diffractive beams is shown in Figure 2. The key parameters in the design are the integration of the prism angle, performance parameters for the diffraction light horizontal/vertical emergent spacing, and variable parameter to the top of the prism and grating spacing and the beam radius.

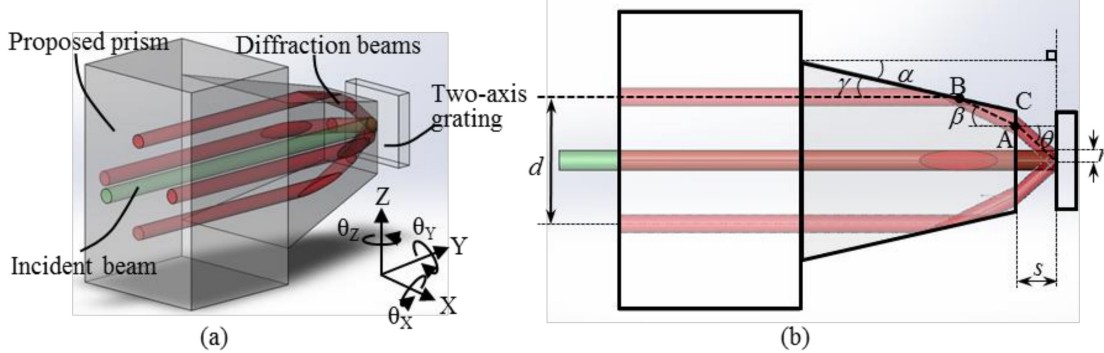

**Figure 2.** (**a**) The 3D structure diagram of the integrated quadrangular frustum pyramid prism and the optical route; (**b**) the optical configure of the defection diffractive beams in the design.

The design and calculation of parameters are described in Figure 2b. The grating diffraction angle is denoted as $\theta$. The refractive angle of diffraction light is denoted as $\beta$. The reflection angle of diffraction light is denoted as $\gamma$. The angle between the normal side of the prism and the top plane is denoted as $\alpha$. The design requires that the diffraction light emitted is parallel to the incident light, then $\alpha$ and $\gamma$ should satisfy Equation (1).

$$\alpha = \gamma. \tag{1}$$

In triangle ABC, there is a geometric relationship expressed by:

$$\gamma + (90° - \beta) + (90° + \alpha) = 180°. \tag{2}$$

Diffraction angle $\theta$ and refractive angle $\beta$ of diffraction light meet the refractive law of Equation (3), where $n$ is the refractive index of optical materials used for prism:

$$\frac{\sin\theta}{\sin\beta} = n. \tag{3}$$

Grating diffraction angle $\theta$ is determined by grating distance d and incident light wavelength $\gamma$:

$$\theta = \arcsin\frac{\lambda}{d}. \tag{4}$$

$$\alpha = \frac{\beta}{2}. \tag{5}$$

$$\alpha = \frac{\beta}{2} = \frac{1}{2}\arcsin\frac{\sin\theta}{n} = \frac{1}{2}\arcsin\frac{\lambda}{nd}. \tag{6}$$

In our design, the optical material of a prism is H-K9Lglass (Union Optic Incorporated, Wuhan, China), and the wavelength $\lambda$ = 660 nm, refractive index $n$ = 1.5128, grating pitch $d$ = 1000 nm. Then, the key parameter of the integrated prism $\alpha$ can be calculated:

$$\alpha = \frac{1}{2}\arcsin\frac{\lambda}{nd} = \frac{1}{2}\arcsin\frac{660}{1.5138 \times 1000} = 12.924°. \tag{7}$$

## 3. Results

Based on the above geometric optical design, a batch of proposed QFP prisms were fabricated by a lens manufacturing company (Union Optic Incorporated, Wuhan, China) and the parameters were measured and given as shown in Figure 3a. It should be noted that a cuboid base was added onto the bottom of the QFP prism so that the prism can be mounted onto the plate base of the grating ruler, which also benefits the prism manufacture process.

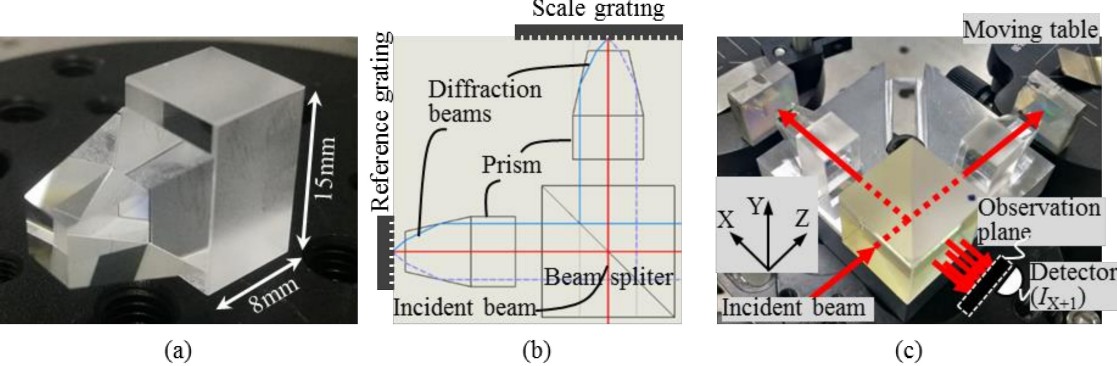

**Figure 3.** (**a**) The picture of the fabricated deflection prism; (**b**) the light path of the deflection of the diffractive beams; (**c**) the grating interferometry testing system with the fabricated deflection prism.

In order to test the effectiveness of the QFP prism, a fundamental grating ruler optical path using interferometric measuring principle is constructed, as shown in Figure 3b,c. This optical layout is the same as the optical path principle shown in Figure 1c. The incident beam with a calibrated wavelength of 660 nm was divided into two beams and are projected to the reference grating and the scale grating after passing through the QFP prisms, respectively. The scale grating is mounted onto a moving table and the reference grating is stable.

First of all, performance of the QFP prism in diffraction beams propagation directions modulation was evaluated by observing the beam spots on the screen. The observation as shown in Figure 3c was placed at a distance about 150 mm, a similar distance with that of a real grating ruler. Figure 4a illustrated the beam spots on the screen. It can be seen that the four diffraction beams from the reference grating coincided well with those from the scale grating. The distance between these four diffraction beams from the central beam (reflective beam, also called zero-order diffraction beam) are consistent with the designed values. These results verified the geometry accuracy of the fabrication process. It should be noted that there are many undemand spots on the screen which are caused by the non-orthogonal diffraction beams from the grating and can be easily blocked by setting a physical holes aperture.

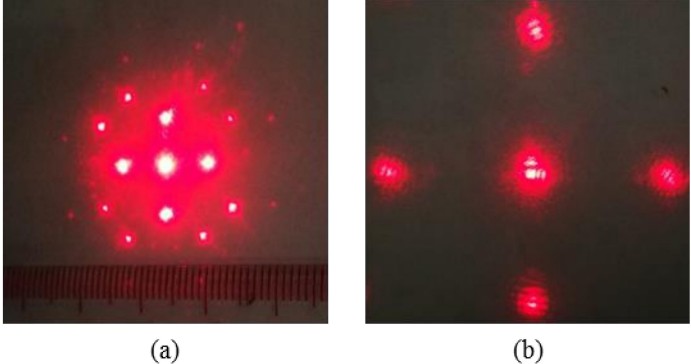

**Figure 4.** (**a**) The diffraction beams spots on the observation screen; (**b**) the interference fringe on the observation screen.

In order to further verify the effectiveness of the proposal in terms of optical property, such as polarization, the interference signal was tested. As shown in Figure 4b, clear diffraction stripe structure can be observed in all four interference spots and these represent that they have almost same order of interference intensity, while in previous separated prism layouts this was always the most challenging task.

The results can preliminarily prove that the QFP prisms based interferometric grating ruler optical configuration is feasible and effective. To further test the optical path, the scale grating was

driven by the micron motion platform (PI-M112.1 DG, Physik Instrumente (PI) GmbH & Co. KG, Karlsruhe, Germany), as shown in Figure 3c.

As the principle of grating interferometry, the movement of the scale grating brings a periodic change of the interference signal. In this demonstration, two interference signals, $I_{X+1}$ and $I_{X-1}$, were recorded by a photodiode. The interference signals are given in Figure 5. It can be seen that the visibility of the interference signal almost remains stable, which is essential for the grating ruler application. The direct current components in the interference signal can be eliminated optically. The results show that the given QFP prisms based optical layout can form a good interference signal, the signal is in good sinusoidal shape under constant speed drive and can be used for bit shifting calculation.

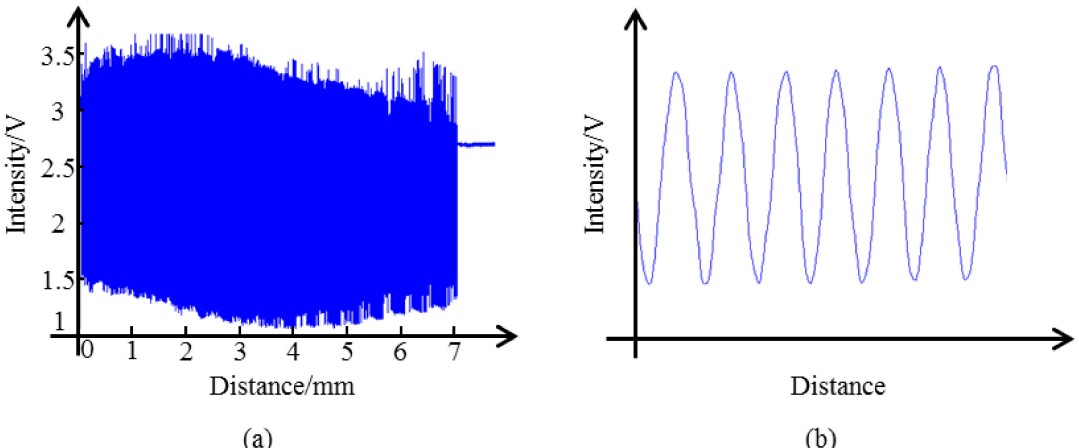

**Figure 5.** (**a**) The interference signal of collected data; (**b**) the detail interference signal of one channel ($I_{X+1}$) during moving.

The motion velocity along the direction X and direction Y of the measured grating was 0.025 mm/s, the motion stroke was 7 mm, and the sampling rate was 1000 Hz. A total of four repeated experiments were conducted. The calculation results of bit shifting of each test are shown in Table 1. As can be seen from the data in the table, the repeatability of the bit transfer calculation is good, and the relative error is no more than 2.5%. The measuring error can be reduced by calibrating the measure system for the machining error in fabricating the grating and the QFP prism and assembly error. In each group, the result of $X_{+1}$ is slight less than $X_{-1}$, because the rear surface of the QFP prism is not fully parallel to the grating surface, which is inevitable and will have no or little influence on the result after calibrating the system and further processing the subdivision signals.

**Table 1.** The result of the displacement solution (μm).

| Testing Group | Result of $X_{+1}$ | Result of $X_{-1}$ |
| :---: | :---: | :---: |
| 1 | 6829 | 6889 |
| 2 | 6843 | 6890 |
| 3 | 6877 | 6891 |
| 4 | 6874 | 6890 |

## 4. Discussion

First of all, the geometrical parameters of the fabricated QFP prisms were evaluated before they were constructed into the optical testing system. The parameters variation is less than 0.1 mm, which proves this structure design and fabrication technology effective. About cost, the price of one QFP prim is about 60 USD when the customized number is no large than 10 in a Chinese optics factory

and this can greatly reduce to be 15 USD when this product number is larger than 1000. The reduction of cost of this QFP prism will be beneficial for the grating ruler application.

Second, because the scope of this research is focused on the design, fabrication, construction and evaluation of this proposed QFP prism, the systematical algorithm for motion determination that was illustrated in References [6–16] was not introduced or not employed to precisely calculate the displacement. Furthermore, because of the slight misalignment of the motion stage and the scale grating, crosstalk will inevitably be involved. The systematical misalignment could be previously calibrated with a compensation factor used [12].

Lastly, it should be also noted that the proposal in this research requires the grating pitch and the light source wavelength to be time-stable or a small order variation within the designed tolerance. Improvement of the tolerance generally can be achieved by enlarging the beam spot, yet will correspondingly enlarge the size of the QFP prism.

## 5. Conclusions

This research proposed, fabricated and tested a new optics, the QFP prism for simultaneously paralleling multiple diffraction beams for grating interferometry application. To get a compact and effective QFP prism we calculated the angle $\alpha$, the key parameter of the prism, according to the wavelength, the reflective index and grating patch, and designed a QFP prism with $\alpha = 12.924°$. The fabricated QFP prism was tested in a grating ruler system to evaluate the prism performance in terms of shape parameters and optical functionality. The testing results preliminary proved that the design and fabrication process are feasible. The less complexity in optical configuration and assembly procedure with usage of the QFP prism will be beneficial for mass production of this kind of grating rulers.

## 6. Patents

A prism for diffraction beams propagation directions modulation, Xinghui Li, Kai Ni, Xiang Xiao, Qian Zhou, Huanhuan Wang, Lijiang Zeng, Weihan Yuan, Xiao Su and Xiaohao Wang, Application No. CN201611217716.2, Open No. CN106646907A, May 10, 2017.

**Author Contributions:** Conceptualization, X.L. and X.X., methodology, X.X., X.L.; validation, Y.S., X.X., and X.L.; resources, Q.Z., K.N., X.L.; data curation, X.X., Y.S., H.L.; writing—original draft preparation, X.L., X.X.; writing—review and editing, X.L., Y.S..; visualization, X.L., X.X.; supervision, Q.Z.; project administration, Q.Z.; funding acquisition, K.N., Q.Z.

**Funding:** This research is supported by the National Natural Science Foundation of China under Project No. 51427805, the Shenzhen fundamental research funding with Grant No. JCYJ20170817160808432 and Grant No. JCYJ20160531195459678, the Youth funding of Shenzhen Graduate of Tsinghua University with Grant No. QN20180003, and the National key research and development program under Grant No. 2016YFF0100704.

**Conflicts of Interest:** The authors declare no conflict of interest.

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
