# Peer review of "Design and Testing of a Compact Optical Prism Module for Multi-Degree-of-Freedom Grating Interferometry Application"

_applsci, doi:10.3390/app8122495_

Reviewer 1 Report

Review:

Manuscript ID: applsci-386099

The aim of the manuscript is to design, to fabricate and to evaluate a key optical component for multi-degree-of-freedom (MDOF) surface encoder. A new lens module is designed for this kind of grating rulers, a single prism called quadrangular frustum pyramid, enabling simultaneously changing the four diffraction beams without any power lose and easy alignment. To test the lens module a grating ruler system is designed and constructed and experimental verification are introduced in the manuscript.

The manuscript is not well written and organized. The introduction must be reviewed. The experimental methodology used is not explained appropriately. The results are not clear. 

Introduction must provide a comprehensive critical review of recent developments in a specific area or theme. Introduction is expected to have an extensive literature review followed by an in-depth and critical analysis of the state of the art. Bibliographic references must be explained individually and not grouped. In this section it would be opportune to introduce the experimental methodology that will be used in the manuscript with respect to the mentioned bibliographic references. Describe how the results will be presented. I suggest add information to better describe what other researchers have done in this area. I recommend rewriting the introduction.

In the section 2. Lines 89-90: “As mentioned in the first section of this paper, Figures 1(b)-(d)”, but the Authors did not mention it in the section 1. Correct it in the manuscript.

In the section 2, lines 132-133: “Diffraction angle θ and refractive angle β of diffraction light meet the refractive law of Equation 2, where n is the refractive index of optical materials used for prism” But in the equation 2 there is not n. Explain it and correct it in the text.

In the same section, is the equation 2 well written? From the figures the sum of α and γ appear is equal to 180º and not β. Explain it….

In the section 3, line139:” Based on the above geometric optical design, a batch of proposed QFP prisms were fabricated”. It would be convenient to explain briefly how it is fabricated. Add this information in the text. 

In the section 4, lines 200-201: “Second, because the scope of this research is focused on the design, fabrication, construction and evaluation of this proposed QFP prism…” The manuscript doesn’t explain how the authors have fabricated and constructed the prism, but only the authors have explained how to design it and how to evaluate its performances. In this sense the manuscript is poor. Add all information about the design, fabrication, construction and evaluation as proposed by the authors.

The conclusions highlighting the reasons for the research, summarizing the steps followed for the development of the results and highlighting the results obtained. Rewrite the conclusions.

Author Response

Thank you very much for your careful review of our paper. Your precious comments and suggestions, which have greatly helped us to improve the paper, are highly appreciated.

The paper has been revised based on the comments and suggestions.

Reviewer 2 Report

Nice paper, good idea, well described, little to comment/criticize.

Just add a horizontal scale to figure 5b

Author Response

(The authors gave the same response as above.)

Reviewer 3 Report

The paper concentrates on the design and testing of a quadrangular frustum pyramid, which forms four parallel diffraction beams. This relatively complex structure constitutes an innovation in context of multi-degree-of-freedom (MDOF) grating interferometry system applied in CNC related issues, which is quite convincingly explained.

While the issue and the optical element itself are interesting and worth being presented in the paper the language is partly rough and not easy to follow. In addition, drawings (especially Fig. 1 a,b,c,d and Fig. 2 b) are not of high quality, which makes them, to some extent, difficult to comprehend.

In lines 131-133, while the equation 2 appears to be correct the text that follows it is confusing. The intention of the authors seems to be clear but the wording is misleading (refractive index -n).

Out of clarity, I would recommend checking out the sequence of the equations and their description.

There is also a mistake in the description of Fig.1.d. (line 113), which is Fig. 1c instead of 1d.

Concluding, I think that the paper needs to be improved since it concerns interesting and useful issues of practical metrology.

Author Response

Thank you very much for your careful review of our paper. Your precious comments and suggestions, which have greatly helped us to improve the paper, are highly appreciated.

The paper has been revised based on the comments and suggestions.

Round  2

Reviewer 1 Report

The authors corrected the manuscript according to the reviewer's suggestions. The manuscript is now well organized and well written. For this I suggest to accept the manuscript in present form for its publication.

Author Response

Dear Reviewer,

Thank you very much for your careful review of our paper. Your precious comments and suggestions, which have greatly helped us to improve the paper, are highly appreciate.

Reviewer 3 Report

The authors have responded sufficiently to all technical questions. There are still some minor language imperfections like:

In line 247: "To get a compact and  effective QFP prism we calculate the angle α, the key parameter of the prism, according the wavelength, the reflective index and grating patch, and design a QFP prism with α=12.924°." the expression "according" should be changed to "according to".

Author Response

Dear Reviewer,

Thank you very much for your careful review of our paper. Your precious comments and suggestions, which have greatly helped us to improve the paper, are highly appreciate. The paper has been revised based on the comments and suggestions.